



# Consistent regional fluxes of CH₄ and CO₂ inferred from GOSAT proxy XCH₄:XCO₂ retrievals, 2010-2014

Liang Feng[1], Paul I. Palmer[1], Hartmut Bösch[2], Robert J. Parker[2], Alex J. Webb[2], Caio S. C. Correia[3],
Nicholas M. Deutscher[4,5], Lucas G. Domingues[3], Dietrich G. Feist[6], Luciana V. Gatti[3], Emanuel Gloor[7],
Frank Hase[8], Rigel Kivi[9], Yi Liu[10], John B. Miller[11,12], Isamu Morino[13], Ralf Sussmann[14], Kimberly
Strong[15], Osamu Uchino[13], Jing Wang[10], Andreas Zahn[16]

1. National Centre for Earth Observation, School of GeoSciences, University of Edinburgh, UK.

2. National Centre for Earth Observation, Department of Physics and Astronomy, University of Leicester, UK.

3 Instituto de Pesquisas Energéticas e Nucleares (IPEN) -Comissao Nacional de Energia Nuclear (CNEN) -Atmospheric Chemistry Laboratory, Cidade Universitaria, Sao Paulo, Brazil.

4. Institute of Environmental Physics, University of Bremen, Germany.

5. Centre for Atmospheric Chemistry, University of Wollongong, Australia.

6. Max Planck Institute for Biogeochemistry, Jena, Germany.

7. School of Geography, University of Leeds, Leeds, UK.

8. Karlsruhe Institute of Technology (KIT), IMK-ASF, 76021 Karlsruhe, Germany.

9. FMI-Arctic Research Center, Sodankylä, Finland.

10. Institute of Atmospheric Physics, Chinese Academy of Sciences, Beijing 100029, China.

11. Global Monitoring Division, Earth System Research Laboratory, National Oceanic and Atmospheric Administration, Boulder, Colorado, USA.

12. Cooperative Institute for Research in Environmental Sciences (CIRES), University of Colorado, Boulder, Colorado, USA.

13. National Institute for Environmental Studies (NIES), Tsukuba, Japan.

14. Karlsruhe Institute of Technology (KIT), Institute of Meteorology and Climate Research - Atmospheric Environmental Research (IMK-IFU), 82467 Garmisch-Partenkirchen, Germany.

15. Department of Physics, University of Toronto, Toronto, M5S 1A7, Canada

16. Karlsruhe Institute of Technology (KIT), Institute of Meteorology and Climate Research (IMK), 76344 Eggenstein-leopoldshafen, Germany



**ABSTRACT**

We use the GEOS-Chem global 3-D model of atmospheric chemistry and transport and an ensemble Kalman filter to simultaneously infer regional fluxes of methane ($CH_4$) and carbon dioxide ($CO_2$) directly from GOSAT retrievals of $XCH_4:XCO_2$, using sparse ground-based $CH_4$ and $CO_2$ mole fraction data to anchor the ratio. This work builds on previously reported theory that takes advantage that:
1) these ratios are less prone to systematic error than either the full physics data products or the proxy $CH_4$ data products; and 2) the resulting $CH_4$ and $CO_2$ fluxes are self-consistent. We show that a posteriori fluxes inferred from the GOSAT data generally outperform the fluxes inferred only from in situ data, as expected. GOSAT $CH_4$ and $CO_2$ fluxes are consistent with global growth rates for $CO_2$ and $CH_4$ reported by NOAA, and with a range of independent data including in particular new profile
measurements (0-7 km) over the Amazon basin that were collected specifically to help validate GOSAT over this geographical region. We find that large-scale multi-year annual a posteriori $CO_2$ fluxes inferred from GOSAT data are similar to those inferred from the in situ surface data but with smaller uncertainties, particularly over the tropics. GOSAT data are consistent with smaller peak-to-peak seasonal amplitudes of $CO_2$ than either a priori or the in situ inversion, particularly over the
tropics and the southern extra-tropics. Over the northern extra-tropics, GOSAT data show larger uptake than the a priori but less than the in situ inversion, resulting in small net emissions over the year. We also find evidence that the carbon balance of tropical South America was perturbed following the droughts of 2010 and 2012 with net annual fluxes not returning to an approximate annual balance until 2013. In contrast, GOSAT data significantly changed the a priori spatial
distribution of $CH_4$ emission with a 40% increase over tropical South America and tropical Asia and smaller decrease over Eurasia and temperate South America. We find no evidence from GOSAT that tropical South American $CH_4$ fluxes were dramatically affected by the two large-scale Amazon droughts. However, we find that GOSAT data are consistent with double seasonal peaks in fluxes that are reproduced over the five years we studied: a small peak in January to April and a larger peak
in June to October, which is likely due to superimposed emissions from different geographical regions.



## 1. **Introduction**


Atmospheric growth of the two most abundant non-condensable greenhouse gases (GHGs), carbon dioxide ($CO_2$) and methane ($CH_4$), increases the absorption of Earth's outgoing infrared radiation (IR) with implications for the radiation budget of Earth's atmosphere and subsequent manifold changes in climate including an increase in global mean temperatures. The most recent international climate agreement aims to limit the rise in global mean temperature to 2 degrees Celsius, which will be attempted by reducing the emissions of human-driven (anthropogenic) GHGs. This approach necessarily assumes we have good knowledge of emissions from all anthropogenic sectors so that targeted reductions are effective. It also implicitly assumes that the Earth's biosphere will continue to be a net annual sink for up to 40-60% of anthropogenic $CO_2$ (e.g., Barlow et al., 2015), and the continued stability of natural reservoirs of $CH_4$. Current scientific knowledge, informed by mostly ground-based data and models, does not confidently support either assumption even on a continental scale. Here, we present the first multi-year record of self-consistent regional net fluxes (sources minus sinks) of $CO_2$ and $CH_4$ inferred from the Japanese Greenhouse gases Observing SATellite (GOSAT). We show these fluxes are significantly different to those inferred from ground-based data, particularly over tropical ecosystems, but are generally consistent with independent data throughout the troposphere.




Inferring $CO_2$ and $CH_4$ fluxes directly from atmospheric observations is an ill-posed inverse problem, with a wide range of scenarios that fit these data. Prior information is used to regularize the problem, with care taken to describe data and prior uncertainties to avoid over- or under-fitting the data. There is a growing and progressive literature on estimating GHG fluxes in which an atmospheric chemistry transport model is used to relate observed atmospheric GHG mole fractions to atmospheric surface exchange fluxes. A number of approaches are used to minimize the model minus observation residual to infer spatial and temporal variations in flux. Errors introduced by the incomplete and uneven coverage of current ground-based observation networks are compounded by atmospheric model errors (e.g., transport and chemistry) resulting in significant discrepancies between flux estimates inferred from different models on spatial scales < O(10,000 km) (e.g. Law et al., 2003; Yuen et al., 2005; Stephens et al., 2007; Peylin et al., 2013).



Space-borne observations of short-wave IR (SWIR) that are sufficient precise to detect small changes in lower tropospheric $CO_2$ and $CH_4$ necessary for flux inference are beginning to improve current understanding of these GHGs. GOSAT (Kuze et al., 2016), launched in 2009, was the first satellite designed purposefully to measure $CO_2$ and $CH_4$ columns using SWIR wavelengths. There is a growing body of literature that has inferred regional $CO_2$ and $CH_4$ fluxes from GOSAT dry-air $CO_2$ ($XCO_2$) and $CH_4$ ($XCH_4$) column mole fractions using the proxy and full-physics data products (Basu et al., 2013; Deng et al., 2013; Houweling et al., 2015; Bergamaschi, et al., 2013; Takagi et al., 2014; Fraser et al., 2014). The resulting flux estimates (particularly for $CO_2$) are often found to be inconsistent with the results based on the surface network, and with each other using different atmospheric transport models or using different versions of retrievals (Chevallier et al., 2014; Houweling et al., 2015). The reliability of the fluxes inferred from GOSAT $XCO_2$ retrievals (Reuter et al., 2014; Feng et al., 2016),





considering bias in current retrievals (Feng et al., 2016) as well as the variations in temporal and spatial coverage (Liu et al, 2015)., is still a subject of ongoing discussions.

We build on previous work that developed a novel approach to estimate simultaneously regional $CO_2$ and $CH_4$ flux estimates from the GOSAT $XCH_4$:$XCO_2$ ratio measurements, which had been until then used exclusively to develop 'proxy' $XCH_4$ retrievals (Fraser et al., 2014). Previous work has

shown that these ratios are less prone to systematic bias that represents a substantial challenge to the full-physics data products. The underlying assumption of the proxy approach is that by taking the ratio of the two retrieved values that have been fitted simultaneously in nearby spectral windows (1.65 μm and 1.61 μm) any interference due to cloud and aerosol scattering will be similar for both retrieved values and will be removed. The ratio is then scaled by a model $XCO_2$ value, under the

assumption that atmospheric gradients of $XCO_2$ are much smaller than $XCH_4$, to generate $XCH_4$ proxy retrievals. Data products generated by the proxy approach are more robust against scattering than the full-physics approach so that there are more usable retrievals over geographical regions that are compromised by seasonal aerosol and cloud distributions, e.g. tropical South America. Fraser et al. (2014) used a series of numerical experiments and the Maximum A posteriori (MAP) approach to

show that these $XCH_4$:$XCO_2$ ratios could be used, in conjunction with in situ observations of $CH_4$ and $CO_2$ mole fractions, to simultaneously estimate regional $CO_2$ and $CH_4$ fluxes. Pandy et al. (2016) used a similar approach but using a 4-D variational assimilation approach to infer $XCO_2$ and $XCH_4$ fluxes for 20 months from April 2009. They found that after correcting biases in the $XCH_4$:$XCO_2$ retrievals, the ratio inversion results in similar agreement with independent $CO_2$ and $CH_4$ observations, as other

inversions based on the in-situ data only or based on individual GOSAT $XCH_4$ and $XCO_2$ products. Here, we use an Ensemble Kalman Filter (EnKF) to assimilate the $XCH_4$:$XCO_2$ ratio data (UoLv6, Parker et al., 2015) from January 2009 to December 2014, inclusive. A comparison between the UoLv6 data set and the ground-based $XCH_4$ and $XCO_2$ data from the Total Carbon Column Observing Network (TCCON) shows a bias of about 0.3%. We use individual in situ and GOSAT observations (instead of

monthly means, Fraser et al, 2014) to estimate monthly fluxes at a higher spatial resolution than Fraser et al., 2014.

In the next section we describe the Ensemble Kalman Filter approach, the observations we use to infer the $CO_2$ and $CH_4$ fluxes and those we use to evaluate the resulting posteriori flux estimates, and a description of the numerical experiments. In section 3 we describe our result, with a particular

focus on tropical South America where we compare our a posteriori model with new aircraft measurements. We conclude the paper in the section 4.

## 2. Methods and Data

### 2.1 Ensemble Kalman Filter

We develop an existing EnKF framework that has been used to estimate $CO_2$ (Feng et al., 2009; 2013;

2016), and $CH_4$ fluxes from the in-situ or space-based measurements of their atmospheric observations (Fraser et al., 2013). In this study, the state vectors are regional fluxes of $CO_2$ and $CH_4$ at location $x$ and time $t$ as:

$$f_p^g(x,t) = f_0^g(x,t) + \sum_i c_i^g BF_i^g(x,t), (1)$$



where $g$ denotes $CO_2$ or $CH_4$ tracer gas and $f_0^g(x,t)$ describes the a priori estimates of $CO_2$ or $CH_4$ fluxes. Following Fraser et al. (2014), our basis function set $BF_i^g(x,t)$ is defined as the pulse-like (monthly) $CO_2$ or $CH_4$ fluxes from different sectors over pre-defined geographic regions. The coefficients $c_i^g$ for both the $CO_2$ and $CH_4$ fluxes form a joint state vector **c** to be estimated by optimally fitting the model to the data.

In the Ensemble Kalman Filter framework, the prior error covariance **P** is represented by an ensemble of perturbations of the coefficients $\Delta \mathbf{C}$: $\mathbf{P} = \Delta\mathbf{C}\Delta\mathbf{C}^T$, where $T$ represents the matrix transpose. The a posteriori coefficient estimates are given by:

$$\mathbf{c}_a = \mathbf{c}_f + \mathbf{K}\left(\boldsymbol{y}_{obs} - H(\mathbf{c}_f)\right), \text{ (2)}$$

where $\mathbf{c}_a$, $\mathbf{c}_f$ are the prior and posterior estimates, respectively; $\boldsymbol{y}_{obs}$ are the observations; and $H$ is the observation operator that relates surface fluxes (i.e., the coefficients) to the observation data (described below), and includes the atmospheric transport model (Fraser et al., 2014).

The Kalman gain matrix **K** in Eq. 2 is approximated by (Feng et al., 2009):

$$\mathbf{K} \approx \Delta\mathbf{C}\Delta\mathbf{Y}^T[\Delta\mathbf{Y}\Delta\mathbf{Y}^T + \mathbf{R}]^{-1}, \text{ (3)}$$

where **R** is the observation error covariance, and $\Delta\mathbf{Y}^T = H(\Delta\mathbf{C})$ projects the flux perturbation (coefficients) ensemble $\Delta\mathbf{C}$ to observation space. We use the GEOS-Chem global 3-D chemistry transport model (v9.02) to relate the fluxes to the observation space. For the experiments reported here we run the CTM model at a horizontal resolution of 4° (latitude) ×5° (longitude), driven by the GEOS-5 (GEOS-FP for 2013 and 2014) meteorological analyses from the Global Modeling and Assimilation Office Global Circulation Model based at NASA Goddard Space Flight Centre. We use monthly 3-D fields of the hydroxyl radical from the GEOS-Chem HOx-NOx-Ox chemistry simulation to describe the main oxidation sink of $CH_4$ (Fraser et al., 2014). We use a four-month moving lag window to reduce the computational costs related to the projection of the perturbation ensemble into the observation space for longer time periods (Feng et al., 2013; 2016)

Where possible we use consistent emission inventories for $CO_2$ and $CH_4$: monthly biomass burning emission (GFEDv4.0, van der Werf et al., 2010) and monthly fossil fuel emissions (ODIAC, Oda and Maksyutov, 2011). To describe atmospheric $CO_2$ variations, we also use monthly-resolved climatological ocean fluxes (Takahashi et al., 2009), and three-hourly terrestrial biosphere fluxes (CASA, Olsen and Randerson, 2004). To describe atmospheric $CH_4$ variations, following Fraser et al. (2014), we use prescribed annual inventories for emissions from oil and gas production, coal mining, ruminant animals (Olivier et al., 2005), termites, and hydrates (Fung et al., 1991). We use monthly-resolved emissions for rice paddies and wetlands for 2009, 2010 and 2011 (Bloom et al., 2012). From January 2012, we fix the rice paddy and wetland emissions to their monthly means between 2009 and 2011. We also include a simple soil sink of $CH_4$ (Fraser et al., 2014).

We define the pulse-like basis functions (Eq. 1) guided by the TransCom-3 regions (Gurney et al., 2002), with each continental region further divided equally into 4 sub-regions. Figure 1 shows the 44 land regions, and 11 ocean regions that we use in this study; in comparison Fraser et al (2014) used 11 land regions and one ocean region. We distinguish $CO_2$ fluxes between four categories: 1) ocean fluxes; 2) anthropogenic emissions; 3). biomass burning; and 4) terrestrial biospheric fluxes. For $CH_4$



fluxes we distinguish between six categories: 1) ocean fluxes; 2) anthropogenic emissions from coal mining; 3) anthropogenic emissions from oil and gas production, fossil fuel combustion, and other; 4) biomass burning; 5) natural fluxes from wetlands, and rice paddies, and; 6) natural fluxes from termites, hydrates and others. In total, we have 143 monthly basis functions for $CO_2$, and 231 monthly basis functions for $CH_4$.

We assume an a priori uncertainty of 60% for the coefficients corresponding to the natural $CO_2$ and $CH_4$ fluxes, and for $CH_4$ emissions from coalmines. We assume an a priori uncertainty of 40% for $CO_2$ anthropogenic emissions, $CO_2$ and $CH_4$ ocean fluxes, and anthropogenic emission of $CH_4$ from the oil and gas industry. We also assume that a priori errors for the same categories are correlated with a spatial correlation length of 800 km, and with a temporal correlation of one month (Feng et al., 2016). We assume that fire emissions of $CO_2$ and $CH_4$ are correlated with a correlation coefficient of 0.5, accounting for the variation and uncertainty of the fire emission factors (Parker et al., 2016).

## 2.2 Observations

We assimilated GOSAT $XCH_4$:$XCO_2$ retrievals and in situ surface observations of $CO_2$ and $CH_4$ mole fraction. We use version 6 of the proxy GOSAT $XCH_4$:$XCO_2$ retrievals from the University of Leicester, UK, including both the nadir observations over lands, and glint observations over oceans. Previous analyses have shown that these retrievals have a bias of 0.3%, with a single sounding precision of about 0.72% (Parker et al., 2015; 2011). In our experiments we globally remove this 0.3% bias from the GOSAT proxy data. We assume that each single GOSAT proxy $XCH_4$:$XCO_2$ ratio retrieval has an uncertainty of 1.2% to account for possible model errors, including the errors in atmospheric chemistry and transport.

We also assimilate $CO_2$ and $CH_4$ mole fraction observations at surface-based sites, which help anchor the GOSAT ratio observations (Fraser et al, 2014). Figure 1 shows the sites we use, from the NOAA observation network (Dlugokencky et al., 2015). We assume uncertainties of 0.5 ppm and 8 ppb for the in situ observations of $CO_2$ and $CH_4$, respectively. We also assume a model error of 1.5 ppm and 15 ppb for $CO_2$ and $CH_4$, respectively. We adopt a larger percentage for the $CH_4$ model error to account for difficulties in modelling chemical sinks of $CH_4$ in atmosphere (Fraser et al., 2013).

To determine the importance of the ratio data we run twin sets of experiments: 1) "ratio" experiment that include the GOSAT data and the in situ data sets, and 2) "in situ" experiment that use only the in situ surface data.

## 2.3 Independent data to evaluate a posteriori estimates

We use independent observations of atmospheric $CO_2$ and $CH_4$ mole fraction to evaluate the atmosphere mole fractions that correspond to the a posteriori fluxes from our inversions. These observations include data collected by TCCON and by four aircraft campaigns. To improve the readability of the main text, we have placed much of the text and many of the Figures associated with the evaluation of the a posteriori fluxes in Appendix A.

TCCON is a global network of ground-based FTS instruments that measure, among other compounds, the total atmospheric columns of $CO_2$ and $CH_4$ (Wunch et al., 2011). We use the bias-corrected TCCON $XCO_2$ and $XCH_4$ data at all available sites from the recent GGG2014 release of the TCCON dataset (Wunch et al., 2015). For a comprehensive description of the network and the available data from each TCCON site, we refer the reader to the TCCON project page (e.g., Feist et



al., 2014; Deutscher et al., 2014; Notholt et al.,2014; Griffith et al.,2014; Iraci et al., 2014; Strong et
al., 2014; Dubey et al, 2014; and Sussmann et al., 2014.).

We also use aircraft measurements from four projects to evaluate our a posteriori model concentrations: 1) Data collected during experiments one through five from the HIAPER Pole-to-Pole Observations (HIPPO) that provide latitude-altitude cross-sections of tropospheric mole fractions of $CO_2$ and $CH_4$ (and other tracers) covering dates during 2009 to 2011 (Wofsy et al. 2012); 2) Data
collected by commercial airliners as part of the Civil Aircraft for the Regular Investigation of the atmosphere Based on an Instrument Container (CARIBIC) experiment, which are mainly at cruise altitudes but also during ascent/descent over airports (Brenninkmeijer et al., 2007; Schuck et al., 2009); 3) Bi-weekly aircraft measurements (surface to 4 km) collected from 2010 to 2012 at four sites over Brazil by IPEN (Instituto de Pesquisas Energeticas e Nucleares) over the Amazon rainforest
(AMAZONICA, Gatti et al. (2014)): Rio Branco (RBA), Tabatinga (TAB), Alta Floresta (ALF) and Santarém (SAN); and 4) aircraft measurements conducted by IPEN for the FAPESP/NERC-funded Amazonian Carbon Observatory (ACO, Webb et al, 2016) close to two of the AMAZONICA sites from 2012 to 2014: Salinópolis (SAH) and Rio Branco (RBH). These two sites were chosen to best represent air before and after travelling across the Amazon Basin. The purpose of these flights was to improve
validation of GOSAT $XCH_4$ and $XCO_2$ data over the Amazon basin so we flew from the surface to 7 km to capture more of the atmospheric column that GOSAT observes. A detailed description of ACO can be found in Webb et al, 2016 and comparison of these data against GOSAT $XCH_4$:$XCO_2$ data are shown below.

## 3 Results

### 3.1 $CO_2$ Fluxes

Figure 2 shows that the in situ only and the ratio inversions result in similar annual net $CO_2$ flux estimates (averaged for 2010 to 2014) over temperate land regions. But compared to the in situ only inversion, the ratio inversion shows a larger net emission over tropical South America, and a smaller net emission from tropical Asia, although the differences are usually within the 1-σ uncertainties.
We also find that the a posteriori fluxes for the ratio inversion generally have smaller uncertainties, in particular over tropical lands.

Figure 3 and Table 1 compare the time series of the prior and posterior global net $CO_2$ flux estimates. They show that global annual a posteriori net flux estimates are 40-60% smaller the a priori estimates (Table 1) due to a smaller net emission during boreal winter and a larger net uptake during
the boreal summer (Figure 3). The corresponding global annual $CO_2$ growth rate agrees with NOAA estimates, inferred from in situ observations, typically within 0.15 ppm/a, except for 2013 when the inversions are 0.3 ppm/a lower than the NOAA-reported value.

Figure 3 also shows that the monthly a posteriori flux estimates by the in situ and ratio inversions are similar over the northern landmasses, with the exception of the summer in 2014 when the ratio
inversion shows significantly smaller uptake. Over the tropical landmasses, a posteriori fluxes from the ratio inversion show a much smaller seasonal cycle, with exception of boreal summer months in 2014 when these fluxes have larger uptake. In general, uncertainties for the monthly fluxes inferred by the ratio inversion (GOSAT + in situ data) are smaller (up to 30%) than using only the in situ data.



This reflects the poor spatial coverage of the current in situ observing network particularly over tropical ecosystems (Figure 1). Over the southern landmasses the a posteriori fluxes for the two inversions are similar and typically within their uncertainties. We find that both inversions show a gradual reduction in the peak-to-trough amplitude, which appears to support a similar downward trend in the a priori estimates from about 9.0 GtC/a between 2010 and 2011 to about 7.5 GtC/a between 2013 and 2014. A posteriori fluxes also consistently show lower net emissions than a priori

values during austral winter months.

## 3.2 CH$_4$ Fluxes

Figure 2 shows that a priori and the a posteriori global annual net CH$_4$ flux estimates are similar (520 Mt/a for the a priori versus 518 Mt/a for the ratio inversion), but their geographical distributions are significantly different. The ratio and in situ only inversions show much larger emissions than the a

priori estimates over tropical lands: by up to 50% larger for tropical South America and for tropical Asia (Figure 2). This increase is partially offset by reduced emissions at mid-latitudes (e.g., temperate South America). Over Eurasia we find that the ratio inversion has 15% smaller emissions than the a priori estimates, but the fluxes inferred from the in situ surface data for the same region are 25% higher than the a priori (Figure 2). Figure 2 also shows that the ratio inversion has much smaller (up

to 60%) uncertainties than the in situ inversions over almost all TransCom land regions, which is due to better spatial observation coverage of GOSAT proxy data.

Figure 4 shows that at the global scale, the monthly a posteriori fluxes inferred from the ratio and in situ inversions have larger seasonal variations than the a priori: a typical seasonal minimum of about 450 Mt/a and a typical maximum of 680 Mt/a, compared to the a priori that has a minimum of 480

Mt/a and a maximum of 620 Mt/a. The larger a posteriori seasonal variation is largely due to the seasonal cycle over northern landmasses that is driven by varying wetland and fire CH$_4$ emissions. The ratio inversions also show a muted peak emission of typically 30 Mt/a during January to February, partially due to peak emissions over southern landmasses during the austral summers. Over northern hemisphere landmasses, the in situ inversion is systematically 5-10% higher than the

ratio inversion from 2010 to 2014. Over the tropics, we find that a posteriori tropical fluxes from the ratio and in situ inversions are generally larger than a priori estimates. Also the ratio a posteriori fluxes are systematically higher than those inferred from the in situ surface data, and show a small upward annual trend (Table 2). Over this region, we also find that the ratio inversion consistently shows a double-peak structure with a small peak between January and April and a larger peak

between June and October (Figure 4). This is not shown by the in situ inversion or by the a priori inventory. A posteriori fluxes for the southern landmasses are generally lower by 30-50 Mt/a than the a priori values, which, together with northern landmasses, partially offsets the increase in tropical CH$_4$ emissions (Figure 4). Over southern hemisphere land masses, the seasonal cycles of the ratio and in situ inversions are similar, although the ratio inversion generally has lower seasonal

minima, with the exception of 2014 when the phase of the ratio inversion is opposite to the in situ inversion.

## 3.3 Model Evaluation

In general the ratio inversion shows the best agreement with independent CH$_4$ observations, particularly over lower latitudes. A posteriori improvements to the CO$_2$ simulation are relatively

small. We find that both the model CO$_2$ and CH$_4$ concentrations reproduce the large-scale spatial (e.g., the North-South gradient) and temporal (seasonal cycle) variations in the HIPPO and CARIBIC



data (section 2.3). The a posteriori simulations reproduce the observed TCCON $XCH_4$ and $XCO_2$ variations. Over most TCCON sites, the a posteriori $XCO_2$ model biases are within 0.8 ppm (< 0.2%), and the standard deviations are smaller than 1.6 ppm. The typical model biases for model $XCH_4$ data are smaller than 10 ppb (i.e., <0.6%), with a standard deviation smaller than 15 ppb. For more details, we refer the reader to Appendix A where we show pictorially the comparisons between observations and the ratio and in situ a posteriori $CO_2$ and $CH_4$ mole fractions.

Here, we focus on tropical South America (Figure 1) for three reasons. First, in situ surface data are particularly sparse over this geographically region, including two sites (Figure 5) over which we use the observed $CO_2$ and $CH_4$ mole fractions to constrain flux estimates: Arembepe, Bahia, Brazil (ABP, -12.770$^o$ latitude, -38.170$^o$ longitude) and Ragged Point, Barbados (RBP, 13.165$^o$ latitude, -59.432$^o$ longitude). Second, they include vulnerable ecosystems that have recently experienced several widespread drought conditions in 2010 and 2012 (see for example, Lewis et al., 2015; Rodrigues and McPhaden, 2014), which have affected their ability of absorbing carbon (Doughty et al., 2015) and increased fire emissions (Gatti et al., 2014; Alden et al., 2016). And third, we report new aircraft profile measurements from the ACO (Webb et al., 2016) that was designed specifically to evaluate GOSAT column observations of $CH_4$ and $CO_2$ (section 3).

Figure 6 shows that the a posteriori monthly $CH_4$ and $CO_2$ flux estimates over tropical South America from the ratio inversion are significantly different from the in situ inversion, as expected given the in situ surface data coverage. However, monthly a posteriori $CO_2$ fluxes from the ratio inversion are not always statistically different from the a priori, reflecting the large a priori uncertainties associated with fluxes over this region. The in situ inversion typically has larger uptake during the dry season (May to September) and smaller emissions during the wet seasons than the ratio inversion. Because the in situ flux estimates over this geographical region rely on observation far away they are particularly sensitive to a priori uncertainties, as expected. We find that assuming a global a priori uncertainty that is 50% smaller than our control run results in an additional net emission of 0.4 GtC/a over tropical South America in 2010. Including the GOSAT ratio data into that sensitivity inversion leads to a smaller net decrease (of 0.13 GtC/a.) in emissions.

Table 3 shows that the a posteriori annual fluxes inferred by the ratio inversion are significantly larger than the in situ inversion in 2010, 2011, and 2012, by about 0.7, 0.4, and 0.5 GtC, respectively. A posteriori fluxes from the ratio inversion shows net emissions are smaller in 2013 and 2014 than in 2010 or 2012, which is due to larger uptake in the dry season and to smaller emissions in the wet seasons (Figure 6). This result reveals the continental-scale impact of the severe droughts in 2010 and 2012 over Tropical Southern America. Our result for 2010 is consistent with recent studies based on regional-scale AMAZONICA aircraft observations (Gatti et al, 2014; van der Laan-Luijkx et al., 2015; Alden et al., 2016). The in situ inversion fails to reproduce this increase in net emissions during 2010 dry season, instead showing a large uptake (Figure 6).

A posteriori $CH_4$ fluxes from the ratio inversion are systematically higher than the in situ inversion (Figure 6). This discrepancy is particularly large during October 2013 to March 2014 when the in situ inversion is lower than typical seasonal values observed during previous years. Figure 5 shows that $XCH_4$:$XCO_2$ ratio measurements over southwest Amazon increase from 4.55 ppb/ppm to about 4.65 ppb/ppm between October-December 2013 and January-March 2014. This is a small but significant





change in the ratio that suggests either enhanced $CH_4$ emissions and/or lower $CO_2$ fluxes. The two
closest in situ sites to the locus of $XCH_4$:$XCO_2$ variability (RPB and ABP) do not reproduce this change.
Consequently, the in situ inversion may not accurately describe these $CH_4$ flux changes over the
continental interior.

Figures 7 and 8 show that a posteriori fluxes from the ratio inversion generally decrease the mean
model difference against independent AMAZONICA and ACO aircraft observations of $CO_2$ and $CH_4$
over the Amazon basin, but with only small improvements to the associated standard deviations. At
some sites the fluxes from the ratio inversion significantly mute the rapid variations in atmospheric
$CO_2$ and $CH_4$ inferred from the in situ data. Figure 7 shows that for $CO_2$ the greatest improvement is
for the central basin sites of RBA+RBH (after 2012), where the bias reduced from -0.62 ppm to 0.01
ppm with accompanying reduction in standard deviation from 3.7 ppm to 2.6 ppm.  We find similar
but smaller reductions at another AMAZONICA site (TBA). Over other AMAZONICA and ACO sides,
the impact of GOSAT $XCH_4$:$XCO_2$ ratios are even smaller. The coarse resolution of our model that
allows us to exploit efficiently the GOSAT and in situ data is one possible explanation for the large
standard deviations (van der Laan-Luijkx et al., 2015; Gatti et al., 2014 ). Figure 8 shows that overall
the ratio inversion better reproduces the AMAZONICA and ACO $CH_4$ data than the in-situ inversion.
The ratio inversion does best at SAN. It also shows a better agreement over RBA as it does for $CO_2$.
After 2012, the ratio inversion shows a positive bias at the two ACO sites SAH and RBH. Assimilating
the $XCH_4$:$XCO_2$ data reduces the standard deviations (by about 4 ppb to 11 ppb) over ALF, TBA and
RBA (RBH after 2012), and slightly (by about 1 ppb) increase the standard deviations at SAN and SAH.

## 4. Summary


Building on previously reported theory, we simultaneously inferred regional $CO_2$ and $CH_4$ fluxes from
the proxy GOSAT $XCH_4$:$XCO_2$ retrievals 2010-2014, inclusive, anchored by geographically sparse in
situ mole fraction data. The main advantage of using these data directly is that the ratio is less
compromised by systematic bias on spatial scales greater than typical model grid resolution (<1000
km) and less than large-scale variations captured by ground-based network (<10,000 km), which
represents a limiting factor to using full-physics $XCO_2$ measurements. Inferring $CO_2$ and $CH_4$ fluxes
together provides a self-consistent methodology.

We showed that a posteriori fluxes inferred from the GOSAT data generally outperformed the fluxes
inferred only from in situ data, as expected given their greater measurement coverage. GOSAT $CH_4$
and $CO_2$ fluxes are consistent with global growth rates for $CO_2$ and $CH_4$ reported by NOAA, and are
generally more consistent than the results based on in situ surface data with a range of independent
data collected throughout the global troposphere (e.g., aircraft profiles and ground-based total
column measurements), and including in particular new profile measurements (0-7 km) over the
Amazon basin that were collected specifically to help validate GOSAT over this geographical region.

We found that large-scale multi-year annual a posteriori $CO_2$ fluxes inferred from GOSAT data are
similar to those inferred from the in situ surface data but with smaller uncertainties, particularly
over the tropics where in situ surface data are sparse. However, we found that GOSAT data are
consistent with smaller peak-to-peak seasonal amplitudes of $CO_2$ than either a priori or the in situ



inversion, particularly over tropical and the southern extra-tropics, where the annual means are similar. Over the northern extra-tropics, GOSAT data infer a larger uptake than supported by the a priori but a smaller uptake than the corresponding in situ data. Using the individual annual means and seasonal variations over 2010-2014, we found evidence from GOSAT that the carbon balance of tropical South America was perturbed following the droughts of 2010 and 2012 when this region was a large annual source of $CO_2$ (0.5-0.6 PgC/a) to the atmosphere, with net annual fluxes not

returning to an approximate annual balance until 2013.

We showed that GOSAT data significantly changed the a priori spatial distribution of $CH_4$ emission with a 40% increase over tropical South America and tropical Asia and smaller (partially compensating) decrease over Eurasia and temperate South America. We find no evidence from GOSAT that tropical South American $CH_4$ fluxes were dramatically affected by the two large-scale

Amazon droughts in 2010 and 2012. However, we reported that GOSAT data are consistent with double seasonal peaks in fluxes that are reproduced over the five years we studied: a small peak in January to April and a larger peak in June to October. Currently, we have no explanation for this phenomena but it is likely due to superimposed emissions from different geographical regions.

Our analysis, in the wider context of other studies, supports the adoption of using space-borne

observations of $CO_2$ and $CH_4$ to better understand the carbon cycle on the continental scale. Well-known weaknesses of these data (e.g., biases in spatial and temporal coverage) can be partially overcome by integrating them with information from other networks and by judicious use of atmospheric chemistry transport models. The next obvious step is to understand how we can improve source attribution of $CO_2$ and $CH_4$ without necessarily resorting to the assumption, as used

here and elsewhere, that a priori fossil fuel emission estimates are correct. Source attribution can be sometimes achieved by exploiting knowledge of spatial distributions of different sources, but techniques that allow more rigorous exploitation of multi-gas correlations must be developed and incorporated into data assimilation systems that will eventually form the backbone to operational systems (e.g., EU Copernicus Atmospheric Monitoring Service to atmospheric $CO_2$).

**Author contributions**

L. Feng and P. I. Palmer designed the experiments and wrote the paper; H. Bösch, R. J. Parker, and Alex Webb provided the GOSAT $XCO_2$ and $XCH_4$ data; N. M. Deutscher, D. G. Feist, R. Kivi, I. Morino, O. Uchino, F. Hase, R. Sussmann, and K. Strong provided access to TCCON $XCO_2$ and $XCH_4$ data; A.

Zahn provided access to CARIBIC $CO_2$ and $CH_4$ mole fraction data. L. V. Gatti, E. Gloor, C. S. C. Correia, L. G. Domingues, and J. B. Miller provided access to aircraft data (AMAZONICA and ACO) over the Amazon basin. J. Wang and Y. Liu provided a preliminary evaluation of $CO_2$ and $CH_4$ fluxes over China. All co-authors provided comments and suggestions on the manuscript.

**Acknowledgements**


Work at the University of Edinburgh was partly funded by the NERC National Centre for Earth Observation (NCEO), and the European Space Agency Climate Change Initiative (ESA-CCI). P. I.





Palmer gratefully acknowledges funding from the NCEO and his Royal Society Wolfson Research Merit Award. NCEO and the European Space Agency Climate Change Initiative funded work at the University of Leicester. RJP was also funded by an ESA Living Planet Fellowship. We thank NERC and FAPESP for their joint funding of the Amazonian Carbon Observatory Project (NERC Reference: NE/J016284/1). M. Gloor was financially supported by the NERC consortium grant AMAZONICA (NE/F005806/1) which we also thank for providing access to additional aircraft profiles. The TCCON Network is supported by NASA's Carbon Cycle Science Program through a grant to the California Institute of Technology. The TCCON stations from Bialystok, Orleans and Bremen are supported by the EU projects InGOS and ICOS-INWIRE, and by the Senate of Bremen. TCCON measurements at Eureka were made by the Canadian Network for Detection of Atmospheric Composition Change (CANDAC) with additional support from the Canadian Space Agency. Works by J. Wang and Y. Liu are funded by Helmholtz-CAS Joint Research Groups (HCJRG-307). We also thank the HIPPO team for their observations used in our model evaluation. We thank G. J. Collatz and S. R. Kawa for providing NASA Carbon Monitoring System Land Surface Carbon Flux Products: http://nacp-files.nacarbon.org/nacp-kawa-01/.



## Appendix A: Wider Geographical Model Evaluation

We use independent observations to evaluate the a posteriori model concentrations that correspond to the flux estimates, acknowledging limitations associated with sparse observation coverage and atmospheric transport model errors (Chevallier et al., 2014). We sample the GEOS-Chem atmospheric chemistry transport at the time and location of each individual observation.

### HIPPO

Figures A1 and A2 show that the ratio inversion is marginally more consistent with HIPPO $XCO_2$ data than the in situ inversion but the spatial error structure are qualitatively similar. The ratio inversion has a positive bias of 0.2 pm and standard deviation of 1.3 ppm compared to the in situ inversion that has a positive bias of 0.3 ppm and standard deviation of 1.3 ppm. The largest standard deviations (up to 0.8%) reflect the ability of models to reproduce small-scale variations, particularly

at the lowest (the planet boundary layer) and the highest (the upper troposphere and lower stratosphere) altitudes. We find small differences (generally within 1 ppm) below 4-6 km between 40°S and 40°N, and much larger differences (up to 2 ppm) in the upper troposphere and in the lower stratosphere north of 45°N.

The ratio and in situ inversions show similar spatial structure to HIPPO $XCH_4$ data. We find a small

negative bias (0-15 ppb) in the middle and lower troposphere between 40°S and 40°N and a larger positive bias (by over 20 ppb) in the extratropical upper troposphere/lower stratosphere. We find the largest discrepancies between model and observed $XCH_4$ in the higher latitude lower stratosphere, in agreement with previous studies (e.g., Alexe et al., 2015 and Pandy et al., 2016) that is mainly due to difficulties in modelling stratospheric chemical processes. As a result, the ratio

inversion and the in situ inversions have similar biases of 0.6 ppb and 0.1 ppb, respectively, as well as similar standard deviations of 27.7 ppb versus 27.5 ppb respectively.

Figure A2 shows that the two a posteriori models reproduce the hemispheric $CO_2$ gradient, typical for boreal spring months, observed by HIPPO-3 experiment. Compared to the in situ inversion, the ratio inversion has a larger negative bias (-0.8 ppm versus -0.4 ppm) around 20°N, in contrast to a

slightly larger positive bias over most of the southern hemisphere. We find that the overall model bias and associated standard deviation of the gridded partial $CO_2$ columns are very small (biases <0.01 ppm and standard deviation < 0.6 ppm). Figure A2 shows that the two a posteriori models also reproduce the hemispheric $CH_4$ gradient observed by the HIPPO-3 experiment. Compared to the in situ inversion the proxy GOSAT $XCH_4{:}XCO_2$ data significantly reduces the negative bias of the $CH_4$

concentrations (by up to 10 ppb) over the tropical regions. The overall bias for the gridded $CH_4$ partial columns is reduced from -5.6 ppb for the in-situ inversion to the -1.5 ppb for the ratio inversion.

### CARIBIC

Figure A3 shows that the two a posteriori models reproduce the observed annual trend of $CO_2$

monthly means and the observed seasonal cycle with smaller amplitude. Underestimation of the seasonal cycle of the upper tropospheric $CO_2$ concentrations is well documented, and believed to be caused by a deficiency in modelling vertical transport (Stephens et al., 2007). Figure A3 also shows that the a posteriori models reproduce the observed trend and seasonal variation of atmospheric



$CH_4$ in the tropical middle/upper troposphere. The ratio inversion has a smaller bias (-0.37 ppb) than
the in situ inversion (-8.27 ppb) but has only modestly improved the associated standard deviation
by 15% from 7.55 ppb to 6.48 ppb.

### TCCON

Figure A4 shows that the two a posteriori models have a similar level of agreement with 15
independent TCCON $XCO_2$ retrievals. For most of these sites, the model $XCO_2$ bias is well within 1.0
ppm, and the standard deviation is between 0.6 and 1.5 ppm. The two exceptions are sites around
Los Angeles, USA: cj ($34.1^o$N, $118.1^o$W) and jf ($34.2^o$N, $118.2^o$W), where the models underestimate
atmospheric $XCO_2$ by 1.5-2.0 ppm, which we attribute to our coarse model resolution. Figure A4 also
shows that assimilating GOSAT $XCH_4$:$XCO_2$ proxy data significantly reduces the model $XCH_4$ bias by up
to 10 ppb over low-latitude TCCON sites. The GOSAT data also helps to reduce the standard
deviations over most of the 15 sites.




## Tables

**Table 1**: A priori and a posterior estimates of the annual net $CO_2$ fluxes for 2010 to 2014 for the global and three contributing regions:  1) north landmasses; 2) tropical landmasses; and 3) south landmasses.  One-sigma uncertainties are given in the brackets.

| Region | Estimate | 2010 GtC/a | 2011 GtC/a | 2012 GtC/a | 2013 GtC/a | 2014 GtC/a |
|---|---|---|---|---|---|---|
| Global | Prior | 8.64(1.64) | 7.52(1.76) | 8.72(1.57) | 7.97(1.63) | 8.10(1.64) |
| | in-situ | 4.83(0.37) | 3.54(0.35) | 5.10(0.34) | 4.61(0.34) | 4.14(0.36) |
| | ratio | 4.87(0.25) | 3.43(0.25) | 5.08(0.24) | 4.66(0.24) | 4.15(0.26) |
| North lands | Prior | 6.63(1.47) | 6.81(1.60) | 7.52(1.44) | 7.51(1.48) | 7.2 (1.53) |
| | in-situ | 4.60(0.15) | 4.47(0.14) | 5.07(0.15) | 4.89(0.14) | 4.90(0.15) |
| | Ratio | 4.68(0.11) | 4.81(0.11) | 5.38(0.11) | 5.05(0.11) | 5.30(0.11) |
| Tropical lands | Prior | 2.57(0.44) | 1.55(0.46) | 1.95(0.38) | 1.53(0.44) | 1.76(0.43) |
| | in-situ | 1.31(0.28) | 0.70(0.29) | 1.08(0.26) | 1.22(0.27) | 1.04(0.27) |
| | Ratio | 1.63(0.18) | 0.59(0.18) | 1.00(0.17) | 1.21(0.18) | 1.03(0.19) |
| South Lands | Prior | 0.84(0.57) | 0.56(0.57) | 0.64(0.49) | 0.32(0.56) | 0.53(0.45) |
| | in-situ | 0.03(0.25) | -0.50(0.25) | 0.15(0.22) | -0.27 (0.23) | -0.38 (0.24) |
| | Ratio | 0.09(0.15) | -0.56 (0.16) | 0.06 (0.15) | -0.31 (0.16) | -0.52 (0.16) |

**Table 2**: The same as Table 1 but for $CH_4$ fluxes.

| Region | Estimate | 2010 Mt/a | 2011 Mt/a | 2012 Mt/a | 2013 Mt/a | 2014 Mt/a |
|---|---|---|---|---|---|---|
| Global | Prior | 519.3 (59.9) | 517.1 (58.5) | 521.1 (58.7) | 521.1 (58.7) | 521.1 (58.7) |
| | in-situ | 524.8 (23.9) | 509.8 (25.2) | 513.9 (24.8) | 509.3 (24.3) | 529.2 (24.2) |
| | Ratio | 521.2 ( 6.2) | 508.1 ( 6.5) | 508.4 ( 6.3) | 514.8 ( 5.9) | 527.8 ( 7.1) |
| North lands | Prior | 250.3 (36.4) | 253.4 (36.6) | 256.2 (36.9) | 256.2 (36.9) | 256.2 (36.9) |
| | In-situ | 262.6 (14.4) | 272.3 (16.5) | 270.9 (16.4) | 269.8 (15.8) | 277.0 (14.5) |
| | Ratio | 230.4 ( 4.4) | 219.2 ( 4.5) | 227.7 ( 4.5) | 226.8 ( 4.3) | 227.8 ( 4.7) |
| Tropical | Prior | 132.3 (25.9) | 128.4 (24.1) | 129.2 (24.2) | 129.2 (24.2) | 129.2 (24.2) |
| | In-situ | 156.4 (15.7) | 146.2 (15.3) | 147.2 (15.6) | 142.4 (15.7) | 147.8 (15.2) |
| | Ratio | 198.0 ( 5.8) | 203.3 ( 5.8) | 200.1 ( 5.7) | 207.1 ( 5.2) | 207.3 ( 5.9) |
| South lands | Prior | 115.4 (26.7) | 114.1 (26.2) | 114.3 (26.1) | 114.3 (26.1) | 114.3 (26.1) |
| | In-situ | 84.3 (11.6) | 70.1 (11.8) | 74.5 (10.8) | 75.8 (10.8) | 83.0 (11.8) |
| | Ratio | 68.1 ( 4.5) | 61.0 ( 4.6) | 56.5 ( 4.3) | 56.3 ( 4.2) | 67.5 ( 4.9) |

**Table 3**:  Same as Table 1 but for $CH_4$ and $CO_2$ fluxes over tropical South America.

| | | 2010 | 2011 | 2012 | 2013 | 2014 |
|---|---|---|---|---|---|---|
| $CO_2$ (GtC/a) | Prior | 0.93 (0.36) | 0.56 (0.40) | 0.53 (0.32) | 0.37 (0.34) | 0.41 (0.37) |
| | Insitu | -0.09 (0.23) | -0.05 (0.25) | -0.01 (0.22) | 0.18 (0.22) | -0.21 (0.23) |
| | Ratio | 0.63 (0.13) | 0.34 (0.14) | 0.53 (0.13) | 0.05 (0.13) | 0.07 (0.14) |
| $CH_4$ (Mt/a) | Prior | 44.1 (18.4) | 40.3 (16.4) | 40.2 (16.4) | 40.2 (16.4) | 40.2 (16.4) |
| | Insitu | 67.0 (11.6) | 59.5 (11.3) | 54.6 (11.6) | 52.9 (11.9) | 59.5 (11.2) |
| | Ratio | 74.4 ( 3.6) | 78.6 ( 3.8) | 74.0 ( 3.5) | 73.4 ( 3.2) | 73.1 ( 3.9) |





# Figures

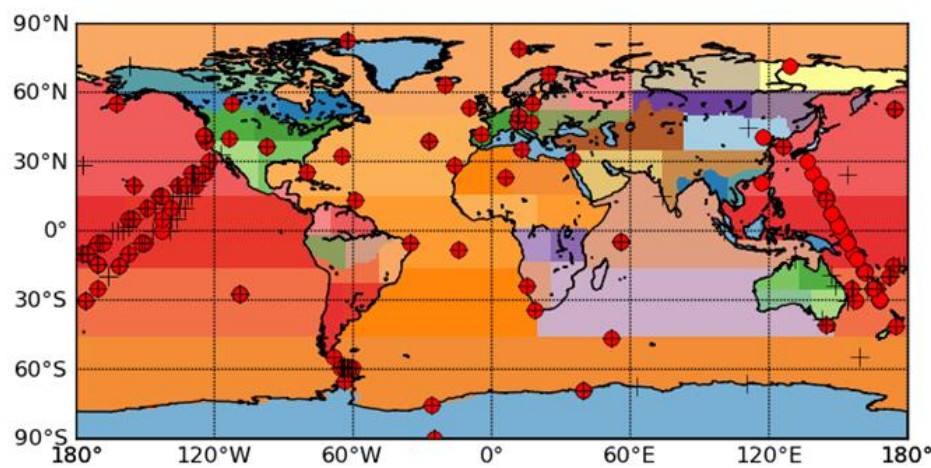

**Figure 1.** Geographic basis functions used in our $CO_2$ and $CH_4$ flux inversion experiments. There are 44 land and 11 ocean regions. The red dots and the black crosses represent the locations of the NOAA in situ $CO_2$ and $CH_4$ observations we assimilate in both the ratio inversion, and the in situ only inversion. Geographical regions are based on those used by the TransCom experiments (Gurney et al., 2002).






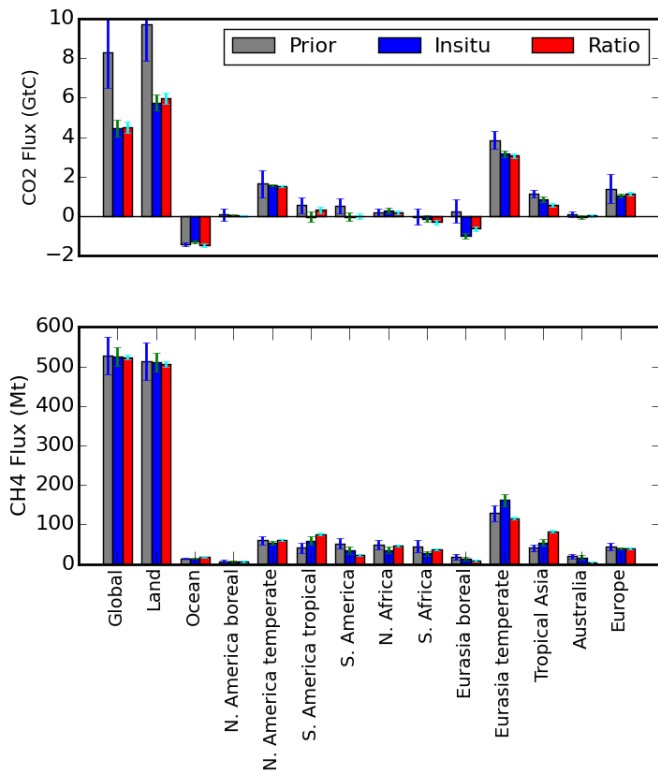

**Figure 2**: Annual mean (2010-2014, inclusive) regional net fluxes of (top) $CO_2$ and (bottom) $CH_4$ inferred from the (red) ratio experiments and the (green) in situ experiments. The grey columns represent the a priori estimates and the vertical lines superimposed on the columns denote one-sigma error. Geographical regions are as defined in Figure 1.


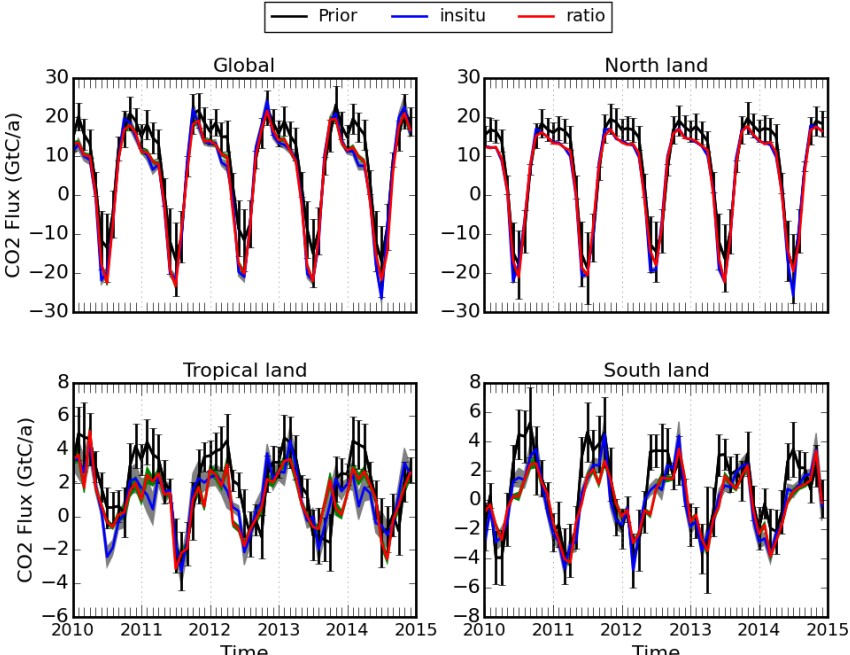

**Figure 3:** The net monthly $CO_2$ fluxes inferred by the in-situ only inversion (blue) and the ratio inversion (red), compared to the prior estimates (black). The vertical lines (envelopes) represent the prior (posterior) uncertainties. In the plots, we aggregate $CO_2$ fluxes of all 4 categories to the net monthly values over 4 predefined global regions (Chevallier et al., 2014): a) global; b) northern lands; c) tropical lands, and; d) south lands.







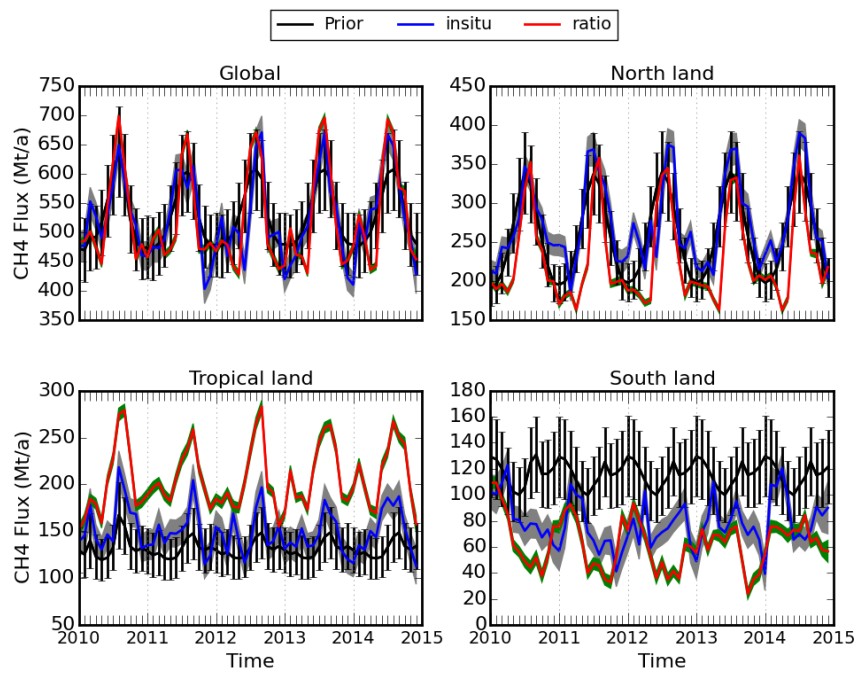

**Figure 4:** As Figure 3 but for CH$_4$ fluxes.




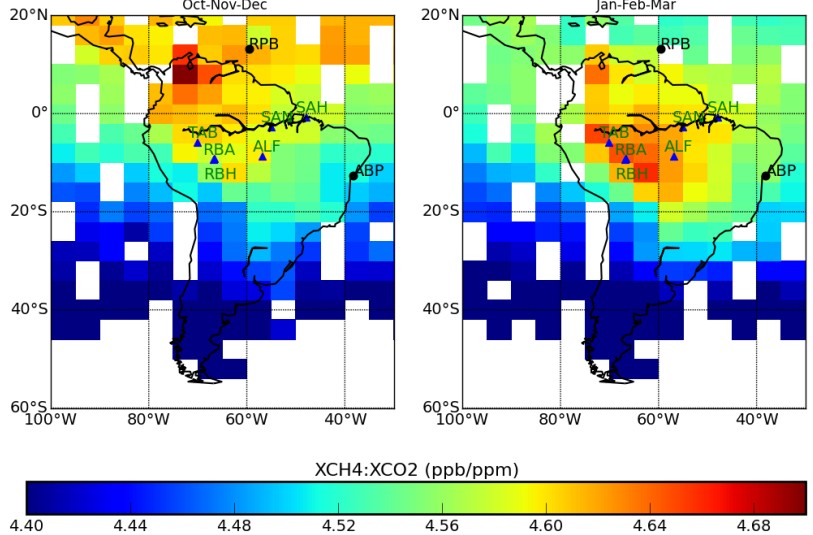

**Figure 5:** GOSAT $XCH_4$:$XCO_2$ ratios over tropical South America (Figure 1) described on the GEOS-Chem $4°$ (latitude) by $5°$ (longitude) averaged over (left) October to December 2013, inclusive, and (right) January to March 2014, inclusive. Black dots represent two NOAA in situ sites RPB and ABP, and triangles represent independent AMAZONICA sites (RBA, ALF, TAB, SAN) and two ACO sites (RBH, SAH), which are described in the main text.





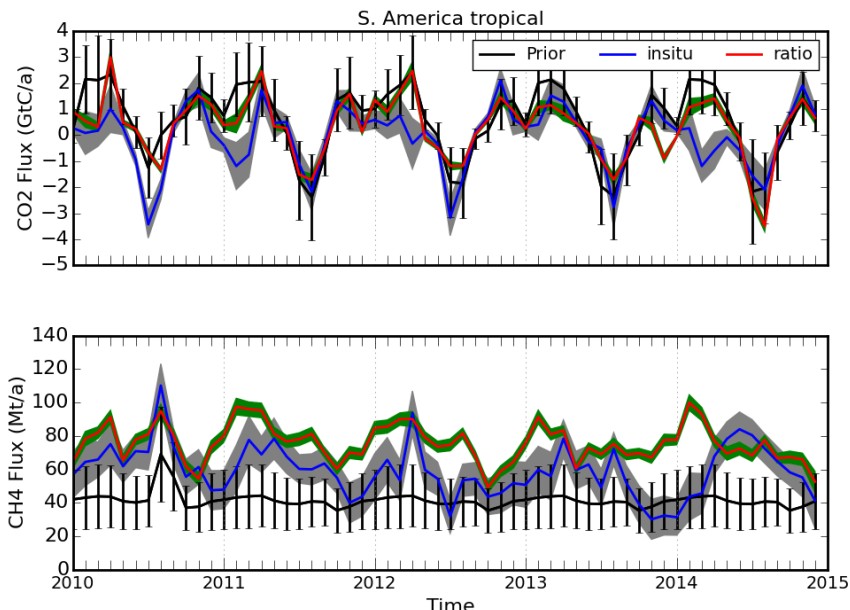

**Figure 6**: As Figure 3 but for $CO_2$ and $CH_4$ fluxes over tropical South America (Figure 1).





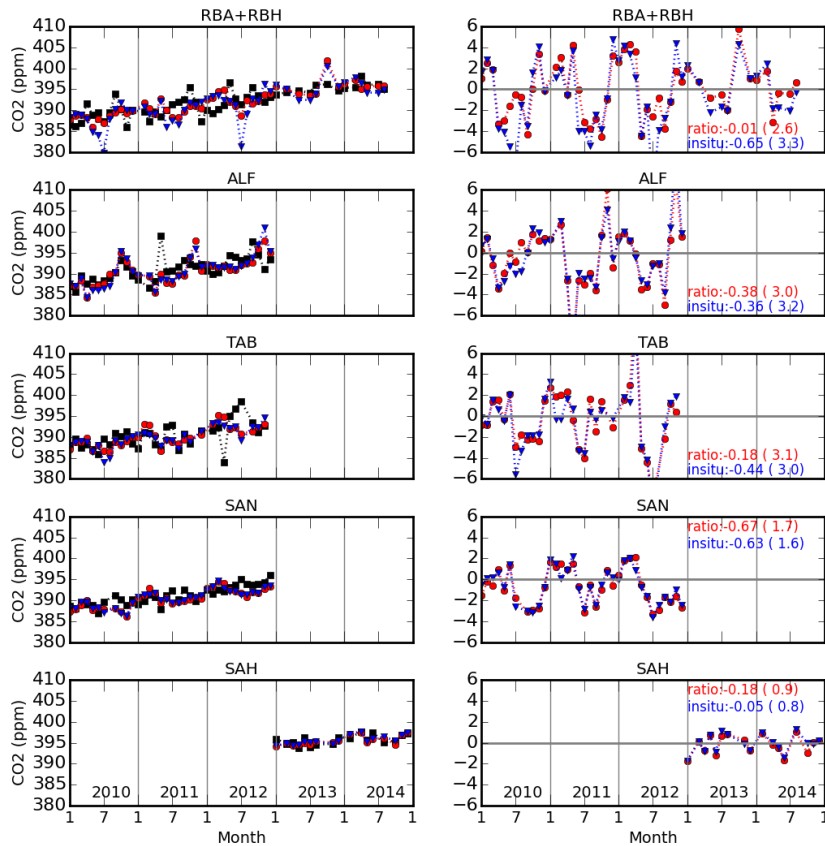

**Figure 7:** Monthly mean partial $CO_2$ columns at four sites over the Amazon (RBA, ALF, TBA, and RBH, Figure 1) collected by the AMAZONICA project and two sites (RBH and SAH) after 2012 collected by the ACO project: (left) comparison and (right) differences with the GEOS-Chem model that has been sampled at the time and location of each observation and driven by fluxes inferred from the in situ (blue) and ratio (red) inversions. The mean and standard deviations (ppm) are shown inset of the right hand side panels. In the plot, we have combined the data over the AMAZONICA site RBA (for 2010 to 2012) and the ACO site of RBH (for 2012 to 2014) for a complete time-series from 2010 to 2014 over the same location.





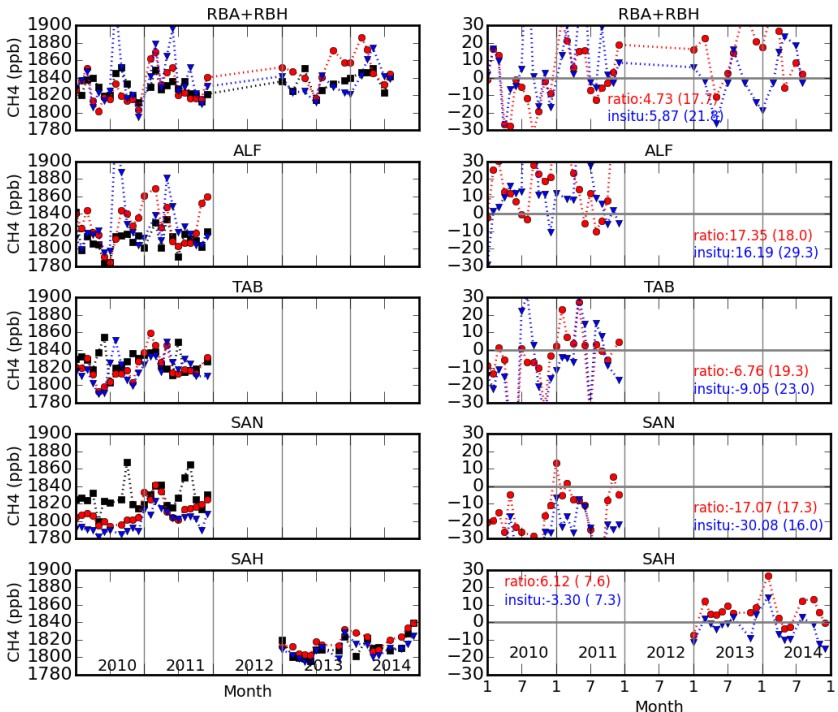


**Figure 8:** As Figure 7 but for comparison of the monthly mean partial CH$_4$ columns (in ppb) of the model simulations with AMAZONICA and ACO observations. Due to availability, CH$_4$ observations for 2012 have not been included.






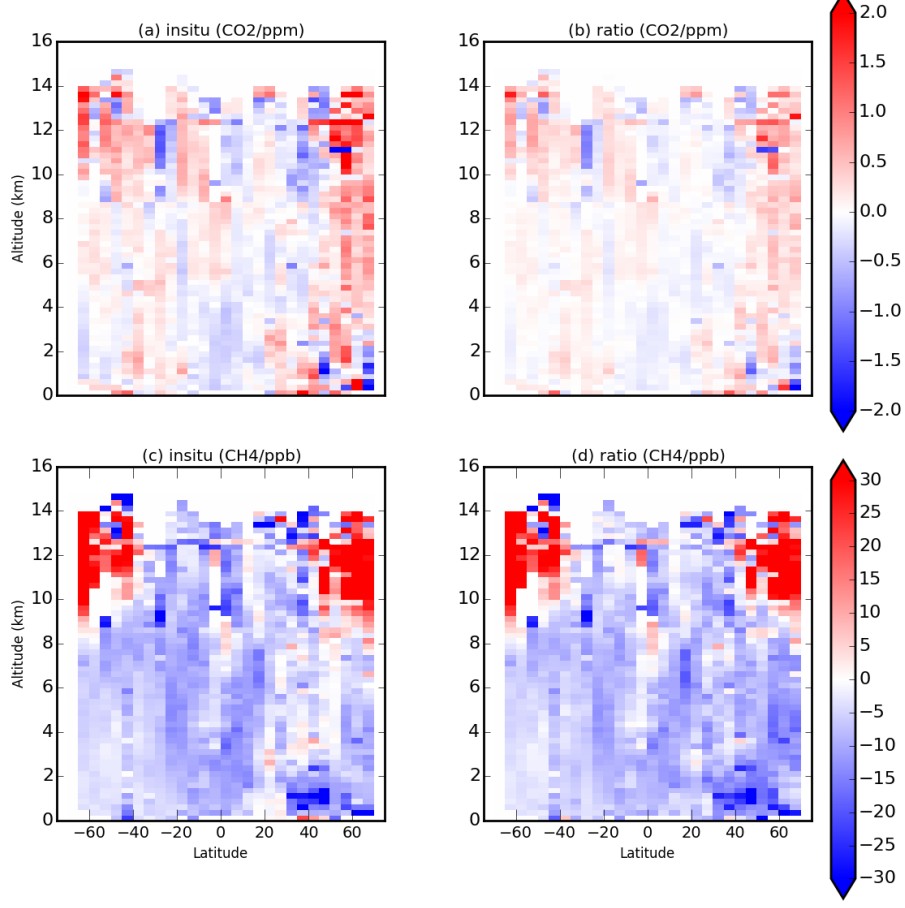

**Figure A1:** Differences between observed and (left) in situ and (right) ratio a posteriori model (top) CO$_2$ and (bottom) CH$_4$ mole fractions observed during HIPPO experiments 1-5 (Wofsy et al., 2011) that cover individual periods during 2009, 2010, and 2011. Model and observation are gridded on a latitude interval of 5 degrees and a vertical interval of 500 m.



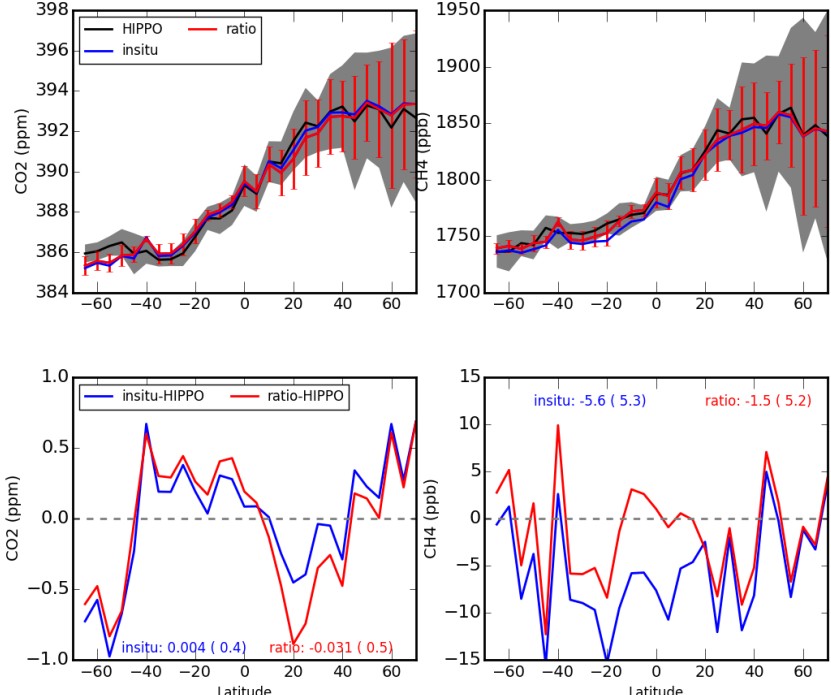

**Figure A2:** (Top) HIPPO-3 (Wofsy et al., 2011), May 2010, and a posteriori model partial columns of (left) CO₂ and (right) CH₄ as a function of latitude over the Pacific Ocean, and (bottom) the differences between the observations and the in situ and ratio inversions. The mean biases (standard deviations) between the model and data are shown in set of lower panels. Data and model values are binned into 5° mass-weighted latitude boxes.






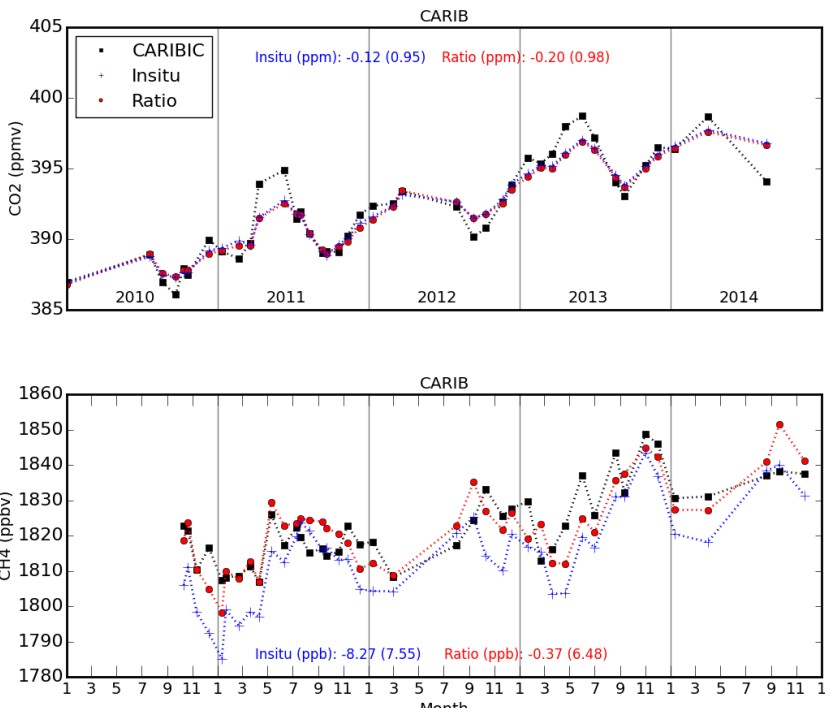

**Figure A3**: Monthly means CARIBIC and a posteriori model (left) $CO_2$ and (right) $CH_4$ mole fractions collected in the tropical middle/upper troposphere (<300 hPa) between 30°S and 30°N. The monthly mean biases (standard deviations) of the model minus data differences are shown inset.




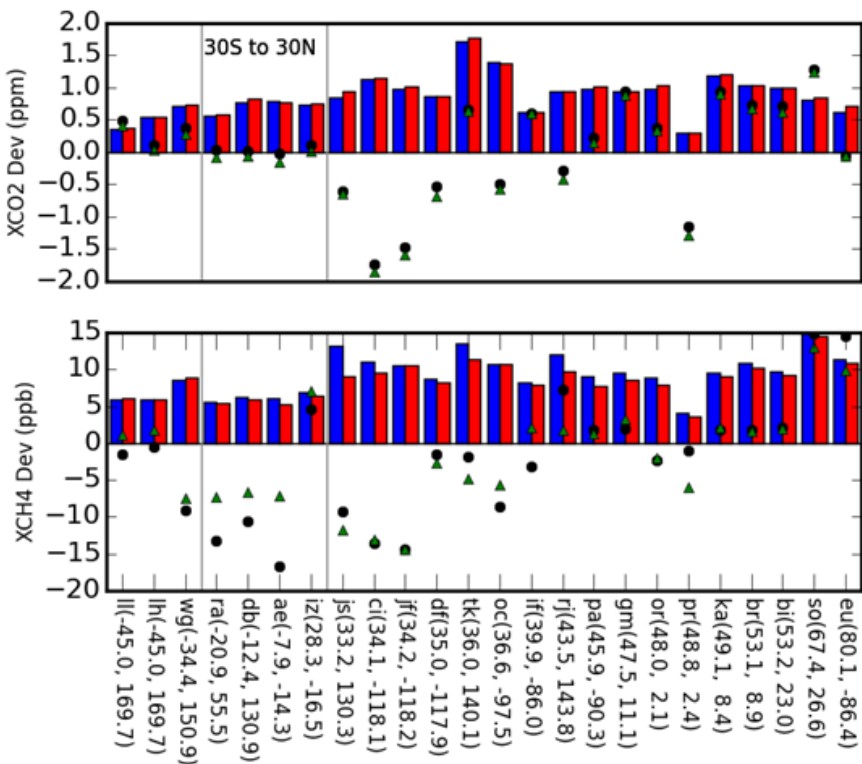

**Figure A4**: Mean multi-year statistics (2010-2014) of the differences between TCCON (top) XCO$_2$ and (bottom) XCH$_4$ measurements and the a posteriori models. Blue and red bars denote the standard deviations between TCCON and the in situ and ratio a posteriori model, respectively. Black circles and green triangles denote the mean deviations TCCON and the in situ and ratio a posteriori models.




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
