# Peer review of "Consistent regional fluxes of CH4 and CO2 inferred from GOSAT proxy XCH4:XCO2 retrievals, 2010-2014"

_Atmospheric Chemistry and Physics, 2016_

## Short Comment (SC1) · 15 Nov 2016

Dear Drs. Feng and Palmer,

A Google Scholar search that I have running (for HIPPO) alerted me to your new discussion paper "Consistent regional fluxes of CH4 and CO2 inferred from GOSAT proxy XCH4:XCO2 retrievals, 2010–2014 ".

I'm commenting to clarify the source of the HIPPO data that you used for your paper.

I work for CDIAC at the Oak Ridge National Laboratory and support the HIPPO Data Archive (http://hippo.ornl.gov/). We watch for data citations to identify papers that have used HIPPO data as we compile our Data Center usage statistics.

[Figure]

In checking your new paper, I see that you appropriately referenced the general Wofsy et al. (2011) paper, but only acknowledge the data product – "We also thank the HIPPO team for their observations used in our model evaluation."

It would appear that the data used was the same as the HIPPO Merged 10-second Meteorology, Atmospheric Chemistry, Aerosol Data (R_20121129).

»> Alternatively, if you used data files (1-second data?) downloaded from EOL, they also provide DOIs for their data products.

If this is correct, please add the full citation with the DOI for the HIPPO Merged 10-second data product.

Wofsy, S. C., B. C. Daube, R. Jimenez, E. Kort, J. V. Pittman, S. Park, R. Commane, B. Xiang, G. Santoni, D. Jacob, J. Fisher, C. Pickett-Heaps, H. Wang, K. Wecht, Q.-Q. Wang, B. B. Stephens, S. Shertz, A.S. Watt, P. Romashkin, T. Campos, J. Haggerty, W. A. Cooper, D. Rogers, S. Beaton, R. Hendershot, J. W. Elkins, D. W. Fahey, R. S. Gao, F. Moore, S. A. Montzka, J. P. Schwarz, A. E. Perring, D. Hurst, B. R. Miller, C. Sweeney, S. Oltmans, D. Nance, E. Hintsa, G. Dutton, L. A. Watts, J. R. Spackman, K. H. Rosenlof, E. A. Ray, B. Hall, M. A. Zondlo, M. Diao, R. Keeling, J. Bent, E. L. Atlas, R. Lueb, M. J. Mahoney. 2012. HIPPO Merged 10-second Meteorology, Atmospheric Chemistry, Aerosol Data (R_20121129). Carbon Dioxide Information Analysis Center, Oak Ridge National Laboratory, Oak Ridge, Tennessee, U.S.A. http://dx.doi.org/10.3334/CDIAC/hippo_010 (Release 20121129)

Don't hesitate to contact me if you have any questions.
* * *

---

## Referee Comment (RC1) · Anonymous Referee #1 · 21 Nov 2016

General comment: The paper is of great importance. It provides a useful method to infer regional flux of CH4 and CO2 based on GOSAT XCH4:XCO2 combined with the ground- based observations. The result is convincing that the ratio-based posterior estimation reduced the uncertainties compared to flux only inferred from in-situ. The authors then discuss the regional differences. Although it is certainly not a surprise the largest differences found in the regions with sparse measurements, e.g. tropics and tropical south America, it is very interesting that a clear increase of CH4 emission in the tropical land and the decrease in the south land is shown in the GOSAT version. Overall, this paper is well written. Its tropic and quality is suitable for publication in the journal. I only recommend that the authors to consider the minor comments and

suggestions as below.

Specific Comments:

1) Please comment on the how sensitive the inversion results are to the transport and chemical scheme of the model? I suppose that the results shown here is under the assumption that the transport of model brings fewer uncertainties compared to the uncertainties of input flux. But I think more cautions should be taken for this assumption.

2) Please comment on how much the results could change when a different region definition is used, e.g. the comparison of the results based on TransCom regions and higher-resolved region definition in this study. I guess the result could be sensitive to how much ground sites are included in each region. So a coarse definition of the regions perhaps favors the in-situ results more than ratio results, is that true?

3) The authors explained and showed in Figure 1 how they defined the regions for the Basis Function. But for the discussion of the results, North lands, Tropical lands and South lands are used. Although they referred the definition to Chevallier et al. 2014, it will be better to show the regions on the map in Figure 1, for example, overlaid on the basis function regions.

4) Line 56 and 400, the name of the region with 'temperate' is wrongly typed as 'temperature'.

5) Line 278, the region should be 'Eurasian temperate' for more accuracy. Please comment on the large decrease of CH4 in Eurasian temperate region in the GOSAT inversion. From my point of view, two inversions go to opposite direction only in this region for CH4. And there are only 4 sites for CH4 and 3 sites for CO2 (if I count right in Figure 1). How does the GOSAT XCH4:XCO2 data show (large spatial or temporal variation)? Do you see large annual variability or increase tendency in the GOSAT retrieval or GOSAT ratio-based inversion?

6) In the caption of Figure 2, color for in-situ experiments should be blue but not green.

7) In the first line caption of Figure 7, 'TAB' instead of 'TBA'. The same for line 360 and 368 on page 10.

8) On page 9, line 329-330, please add a comma after 'Because the in situ flux far away'.

---

## Referee Comment (RC2) · Anonymous Referee #2 · 21 Nov 2016

General comments

The paper provides another valuable development in the direction of using XCH$_4$/XCO$_2$ ratio in the inversion of both the CO$_2$ and CH$_4$ fluxes. This paper is one the first few papers on the subject, another one was published by Pandey et al, (2016). In this paper a longer analysis period is used, allowing for more extensive validation. Ability of the XCH$_4$/XCO$_2$ ratio to constrain fluxes of CO$_2$ and CH$_4$ and improve match with independent observations constitute most appealing and encouraging result of this study. The paper is well written and deserves publication with only minor corrections.

Specific comments

[Figure]

Larger tropical $CH_4$ fluxes are inferred with GOSAT ratio as compared to surface data inversion. How to prove that the result is robust with respect to biases in retrieval and even retrieval prior concentration profiles? Another possible suspect could be the transport model bias in the stratosphere for either $CO_2$ or $CH_4$ or both. Can authors add more discussion on this issue?

L178 The benefit of dividing Transcom regions into 4 relatively equal ones was extensively explored by Patra et al (2005).

Suggestions for technical corrections

L058 Better tell which fluxes are being discussed, suggest to change "fluxes" to "Amazonian fluxes", the context is ambiguous here.

L094 Suggest correcting "sufficient" to "sufficiently"

L100 Houweling et al 2015 is referred to, but not found in references.

L121 Suggest correcting Pandy to Pandey

L149 When introducing "prior covariance" need to tell which covariance - fluxes or concentrations?

L396 Text "GOSAT data significantly changed the a priori spatial distribution" should be modified towards saying that posterior changes significantly with respect to prior.

L462 Wording "XCH$_4$ in . . . lower stratosphere" doesn't sound right.

References

Patra, P.K., M. Ishizawa, S. Maksyutov, T. Nakazawa, and G. Inoue, Role of biomass burning and climate anomalies on land-atmosphere carbon fluxes based on inverse modelling of atmospheric $CO_2$, Global Biogeochem. Cycles, 19, GB3005, doi:10.1029/2004GB002258, 2005

---

## Author Comment (AC1) · 20 Dec 2016

The CO2 and CH4 data are extracted from the merged 10-second HIPPO data file, which was downloaded from http://hippo.ornl.gov/ about two years ago. We will update the data reference in the revision.

Regards

Liang Feng
* * *

---

## Author Response (AR1)

**Consistent regional fluxes of CH$_4$ and CO$_2$ inferred from GOSAT proxy XCH$_4$:XCO$_2$ retrievals, 2010-2014**

By Feng et al

We thank both reviewers for their support and comments that we believe have helped to improve the revised manuscript. Below are our responses to each individual reviewer comments (denoted in italics). Changes to the text are shown in red.

**Review report 1**

*General comment:*
*The paper is of great importance. It provides a useful method to infer regional flux of CH4 and CO2 based on GOSAT XCH4:XCO2 combined with the ground- based observations. The result is convincing that the ratio-based posterior estimation reduced the uncertainties compared to flux only inferred from in-situ. The authors then discuss the regional differences. Although it is certainly not a surprise the largest differences found in the regions with sparse measurements, e.g. tropics and tropical south America, it is very interesting that a clear increase of CH4 emission in the tropical land and the decrease in the south land is shown in the GOSAT version. Overall, this paper is well written. Its tropic and quality is suitable for publication in the journal. I only recommend that the authors to consider the minor comments and*

*Specific Comments:*

*1) Please comment on the how sensitive the inversion results are to the transport and chemical scheme of the model? I suppose that the results shown here is under the assumption that the transport of model brings fewer uncertainties compared to the uncertainties of input flux. But I think more cautions should be taken for this assumption.*

The other reviewer also raised this issue.

We agree of course with the reviewers that model error, including atmospheric transport and model chemistry errors, does affect the resulting flux estimates inferred from top-down flux inversions (for example, Chevallier et al., 2010). We also acknowledge the challenge of quantifying the sources of such errors. In our study, we have assumed a simple (and transparent) description of model error and included it as part of the observation errors for the GOSAT XCH4:XCO2 ratio and for the in-situ CH4 and CO2 observations. To address the reviewers' comment we have included in the revised manuscript an explicit acknowledgement of the challenge associated with quantifying model error.

Lines 213-216:

A robust description of model error remains a major challenge for this and similar studies. We have assumed a simple formulation to describe model error, which will not fully account for impacts of errors from, for example, model atmospheric transport on resulting CO2 and CH4 flux estimates.

*2) Please comment on how much the results could change when a different region definition is used, e.g. the comparison of the results based on TransCom regions and higher-resolved region definition in this study. I guess the result could be sensitive to how much ground sites are included in each region. So a coarse definition of the regions perhaps favors the in-situ results more than ratio results, is that true?*

We split the TransCom regions and distinguish between sources of CO$_2$ and CH$_4$ to reduce the impact of aggregation error associated with inversions described at coarse spatial resolutions. In the revised manuscript we explicitly make that point (line 181).

We describe the inversion on these smaller geographic regions to help reduce aggregation errors associated with fluxes being estimated on a coarse spatial resolution (Patra et al., 2005).

We agree that increasing the granularity of the inversion grid could affect the relative importance of the in-situ and satellite observations for constraining flux estimates. Investigating this further is
outside the scope of this study. However, we have added an acknowledgement of this reviewer point in the summary (line 413)

'…the sensitivity of our results to model error and to the temporal and spatial resolution of fluxes requires further investigation, '

As a preliminary assessment to answer the reviewer's question (to satisfy our curiosity but not suitable for publication) we ran another calculation. We artificially enlarged the prescribed errors for in-situ $CO_2$ and $CH_4$ observation by 30% to reduce their relative importance. The results show that
for almost every TransCom land regions, the changes in the resulting $CO_2$ and $CH_4$ flux emissions are small. For example, over tropical South American annual net $CH_4$ emissions for 2010 changed by < 10%, where coverage by in-situ data is very sparse.

*3) The authors explained and showed in Figure 1 how they defined the regions for the Basis Function. But for the discussion of the results, North lands, Tropical lands and South lands are used. Although they referred the definition to Chevallier et al. 2014, it will be better to show the regions on the map in Figure 1, for example, overlaid on the basis function regions.*

Good suggestion. We added another panel to Figure 1 to show the definition of northern, tropical and southern land regions.

*4) Line 56 and 400, the name of the region with 'temperate' is wrongly typed as 'temperature'.*

Typos corrected.

5) *Line 278, the region should be 'Eurasian temperate' for more accuracy. Please comment on the large decrease of $CH_4$ in Eurasian temperate region in the GOSAT inversion. From my point of view, two inversions go to opposite direction only in this region for $CH_4$. And there are only 4 sites for $CH_4$*
*and 3 sites for $CO_2$ (if I count right in Figure 1). How does the GOSAT $XCH_4:XCO_2$ data show (large spatial or temporal variation)? Do you see large annual variability or increase tendency in the GOSAT retrieval or GOSAT ratio-based inversion?*

The GOSAT $XCH_4:XCO_2$ ratios show large spatial variations over Eurasian temperate (see the
Figure below). The available in-situ observations are not sensitive to emissions from southeast China, which have strong $CH_4$ sources from wetlands and agriculture. As a result, assimilating in-situ data only will not correct for any prior overestimation of these $CH_4$ sources. In contrast, the GOSAT ratio data provide useful constraints of emissions for this geographical region. To clarify this point in the revised manuscript, we add (Line 285-287):

', which is due to the in situ network having little sensitivity to emissions over a large part of Eurasian temperate, in particular over southeast China where there are large $CH_4$ sources from wetlands and rice paddies.

We find that the GOSAT $XCH_4:XCO_2$ ratios show a small but consistent downward trend, which is partially due to the increasing $CO_2$ concentrations. The $CH_4$ emission estimates show no consistent trends from 2010 to 2014. However, due to very limited independent observations over the region, we are unable to evaluate the results.

[Figure]

*Figure: Annual mean GOSAT XCH4:XCO2 ratios for 2013, which are gridded into the 4x5 model boxes.*

*6) In the caption of Figure 2, color for in-situ experiments should be blue but not green.*

Typo corrected.

*7) In the first line caption of Figure 7, 'TAB' instead of 'TBA'. The same for line 360 and 368 on page 10.*

Typos corrected.

*8) On page 9, line 329-330, please add a comma after 'Because the in situ flux far away'.*

Typo corrected.

**Review report 2**

**General comments**

The paper provides another valuable development in the direction of using XCH4/XCO2 ratio in the inversion of both the CO2 and CH4 fluxes. This paper is one the first few papers on the subject, another one was published by Pandey et al, (2016). In this paper a longer analysis period is used, allowing for more extensive validation. Ability of the XCH4/XCO2 ratio to constrain fluxes of CO2 and CH4 and improve match with independent observations constitute most appealing and encouraging result of this study. The paper is well written and deserves publication with only minor corrections.

**Specific comments**

*Larger tropical CH4 fluxes are inferred with GOSAT ratio as compared to surface data inversion. How to prove that the result is robust with respect to biases in retrieval and even retrieval prior concentration profiles? Another possible suspect could be the transport model bias in the stratosphere for either CO2 or CH4 or both. Can authors add more discussion on this issue?*

Good question. Other GOSAT CH4 inversions (e.g. Alexe et al., 2015) have also reported elevated CH4 emissions over tropical regions While our posterior fluxes, inferred from the ratio inversion, are in better agreement with independent observations over the tropics than our prior fluxes (Figures A2 to A4), we are unable to exclude the impact of observation biases, and particularly model errors. This is partially due to scarcity of independent data over this region.

In our response to reviewer 1 (above) we have discussed the challenge surrounding robust quantification of model errors that effectively limits our (and everyone else's) ability to assess their impact on flux estimates. We have chosen to explicitly acknowledge this issue to ensure the reader is aware of it (Line 213-216):

A robust description of model error remains a major challenge for this and similar studies. We have assumed a simple formulation to describe model error, which will not fully account for impacts of errors from, for example, model atmospheric transport on resulting CO2 and CH4 flux estimates.

*L178 The benefit of dividing Transcom regions into 4 relatively equal ones was extensively explored by Patra et al (2005).*

Not including this paper was egregious oversight on our part. In the revision, we add (Line 181):

We describe the inversion on these smaller geographic regions to help reduce aggregation errors associated with fluxes being estimated on a coarse spatial resolution (Patra et al., 2005).

See a further discussion of this point in our response to second point from the first reviewer.

**Suggestions for technical corrections**

*L058 Better tell which fluxes are being discussed, suggest to change "fluxes" to "Amazonian fluxes", the context is ambiguous here.*

Done.

*L094 Suggest correcting "sufficient" to "sufficiently"*

Done.

*L100 Houweling et al 2015 is referred to, but not found in references.*

Reference added.

*L121 Suggest correcting Pandy to Pandey*

Typo fixed.

*L149 When introducing "prior covariance" need to tell which covariance - fluxes or concentrations?*

Changed to
'prior flux error covariance'.

*L396 Text "GOSAT data significantly changed the a priori spatial distribution" should be modified towards saying that posterior changes significantly with respect to prior.*

We have changed this to

'GOSAT data results in significant changes with respect to the a priori spatial distribution'

*L462 Wording "XCH₄ in : : : lower stratosphere" doesn't sound right.*

Typo fixed.

*References*

[revised manuscript text omitted]

---

## Author Response (AR2)

We thank Dr. Rolf Müller for his kind help and constructive comments. We have made technical corrections according to the comments:

1) *Update the reference list, e.g. Reuter et al 2014 is still listed as an ACPD paper*

   5    Done

2) *l. 625 phenomena --> phenomenon*

   Done

10   3) *New Fig 1, top: explain the colours in the figure caption.*

   We now add (in Figure 1 caption):

[revised manuscript text omitted]